# Sustainable Development Goals in the Business Sphere: A Bibliometric Review

**Javier Martínez-Falcó** [1,2,*], **Bartolomé Marco-Lajara** [1], **Eduardo Sánchez-García** [1] **and Luis A. Millan-Tudela** [1]

1   Management Department, University of Alicante, 03690 San Vicente del Raspeig, Spain
2   Department of Geography and Environmental Studies, Stellenbosch University, Stellenbosch 7600, South Africa
*   Correspondence: javier.falco@ua.es

**Abstract:** Academic contributions on the impact of the Sustainable Development Goals (SDGs) on businesses have grown exponentially in recent years as a result of the importance of the business sector in improving the economy, society and the environment. Through the use of bibliometric methods and taking the Web of Science (WoS) as a reference database, the research aims to analyze the structure of scientific knowledge of the link between the SDGs and the corporate sector, analyzing 2366 documents published between 1992 and 2022. The results show, among other aspects, the accelerated growth rate of the scientific production analyzed since 2015, the use of publications in articles as the main format for disseminating research results, the relevance of the category of Environmental Sciences as the area of study in which most of the scientific production analyzed falls as well as the predominant role of the publishing houses MDPI, Elsevier and Emerald in the publication of scientific documents on the topic under analysis. The research can therefore be of use to both neophyte and experienced researchers who wish to deepen their understanding of the academic knowledge structure of the SDGs in the business world.

**Keywords:** sustainable development goals; business; bibliometric analysis; VosViewer; Web of Science

## 1. Introduction

The Millennium Development Goals (MDGs) brought together and channeled the efforts of different countries in a movement against poverty, which has so far been the most successful in history [1]. The United Nations Millennium Declaration was signed in September 2000 and committed leaders and their governments to fight poverty, hunger, disease, illiteracy, environmental degradation and discrimination against women [2]. Despite the many gains made, in 2015, United Nations Secretary Ban Ki-Moon warned that he was aware that inequalities persisted and that progress had been uneven, but that further progress required unwavering political will and a long-term collective effort [3].

The MDGs gave way to the Sustainable Development Goals (SDGs). In September 2015, the Heads of State and Government of more than 150 countries, meeting in an Assembly at the United Nations, approved the 2030 Agenda for SDGs, comprising 169 targets, grouped into 17 goals aimed at poverty eradication and sustainable development in its social, economic and environmental dimensions [4]. In this case, not only the participation of all governments was called for, but also the help of citizens and businesses was essential [5]. The United Nations created the Statistical Commission in relation to the 2030 Agenda, with the aim of developing the Global Indicators Framework for the goals and targets of the 2030 Agenda for Sustainable Development [6]. The different SDG signatory countries approved, also at an assembly in 2017, this set of common indicators. Therefore, all countries now have indicators with which to measure the progress made in each territory [7]. Each government must establish strategies and objectives, make investments and measure them [8]. However,

there are no collective or common indicators, neither for the role that citizens can play, nor for the contribution of companies [9].

At a time of crisis resulting from the pandemic situation, companies are increasingly aware of the need to contribute to a more sustainable environment and a fairer society [10]. Investors are also increasingly focusing on the sustainability of the businesses in which they invest, realizing that these companies can better manage risks, identify business opportunities and be resilient to future crises [11]. This increased attention to sustainability has, in turn, been driven by regulators, who are increasingly interested in making companies aware of the impacts they have on their stakeholders [12–18]. In this sense, each type of company and each sector should make an effort to identify the SDGs with which they can interact (those that represent an opportunity for them or have the greatest impact), define an alignment strategy, define clear indicators, measure the results and make them public [19].

The literature related to sustainability at the corporate level is increasingly extensive [20]. Indeed, in the last decade, research on Corporate Social Responsibility (CSR) and Corporate Sustainability (CS) reflects a linkage between these types of activities and aspects such as business performance, risk or profit management [21]. However, while previous findings on CSR and CS activities can be a starting point for the literature on the SDGs, it is important to note that there is a clear difference between the SDGs and CSR and CS, as while the SDGs are more related to global challenges and focus on a macro-level perspective [22], CSR and CS rely on specific strategies with a clear benefit and high impact on the corporation [23]. In this regard, there is previous research that has analyzed the literature on the SDGs at the macro level, as well as studies that address the implementation of sustainability principles at the corporate level. However, as discussed in the next subsection, to the best of our knowledge, there are no up-to-date studies using bibliometric methods to analyze the role of business in the SDGs. In order to overcome this gap in the literature, this research aims to conduct a bibliometric analysis of the literature on the SDGs at the business level.

In this line, this study offers a bibliometric analysis of the literature that has addressed the SDGs and business management jointly, considering all existing publications on the subject in the Web of Science (WOS) database until the year 2022. Using advanced bibliometric techniques and VosViewer software, the aim is to identify the seminal articles, authors, institutions, countries and collaborations between the most important researchers that have laid the foundations and paved the way for research on the study of the SDGs in business. Bibliometric analysis has been chosen over conventional review methods because bibliometric techniques are unbiased, objective, analytical, robust, transparent and valuable in revealing unique networks within a given field of study, as well as providing an overview of the field [24].

In order to achieve the research objective, the research is structured as follows. First, after this brief introduction, Section 2 elaborates on the justification of the research gap, Section 3 presents the methodology, Section 4 shows the results and discusses them and, finally, Section 5 presents the main conclusions, limitations and future lines of research.

## 2. Literature Reviews in the Field of SDGs in the Business Context

As the volume of scientific production on the SDGs at the business level has increased, it has become necessary to collect, classify and analyze the active research fronts around the topic. In this regard, Table 1 shows the publications in journals indexed in the WOS core collection that aim to review the literature on the role of business in the SDGs, classifying the papers by their authors, the journal in which they have been published, the objective of the study, the type of review, the period covered by the review and the number of papers analyzed.

As can be seen, there are only 12 research papers indexed in the main collection of the WOS that have reviewed the literature on the subject under study, which shows the need to continue advancing the state of the art of the discipline. In terms of scientific output,

all articles were published in the last three years (period 2020–2023), which highlights the intensification of the SDG literature at the business level. Furthermore, of the 12 reviews, 8 are bibliometric, 2 are systematic and the remaining 2 use both bibliometric and systematic methods.

In terms of the subject matter of the scientific output identified, only the research by Pizzi et al. [25], Garrido-Ruso et al. [26] and Lee et al. [27] focus specifically on the analysis of scientific literature that has analyzed the SDGs at the business level, given that the rest of the research addresses this linkage but also analyses other business concepts, such as artificial intelligence [28], CSR [29], intellectual capital [30], integrated information [31], CS [32], supply chain [33], digitalization [34], sustainable development [35] and design thinking [36]. Likewise, of the 12 reviews of the literature, only the studies by Ye et al. [29], Meseguer-Sánch et al. [32] and Jan et al. [35] exceeded 1000 documents analyzed.

The reviews of the literature on the topic analyzed allow for identifying a series of shortcomings that need to be addressed. First, despite the growing interest in the role of business in meeting the SDGs, the number of research studies that have reviewed the literature is still scarce, as only 12 reviews have been identified in the main collection of the WOS that address this objective. Second, no review includes the year 2022 in its review period, so reviews can be updated up to the present time. This represents an opportunity to advance scientific knowledge of the discipline given that, as scientific production on the subject under analysis has experienced exponential growth, it needs to be reviewed periodically. Third, the number of publications analyzed in each review is mostly less than a thousand documents, so there is a need to review the literature on the SDGs at the business level more broadly in order to reach a more comprehensive view of the subject matter. In order to overcome the deficiencies detected, the aim is to carry out a review of the literature using bibliometric methods, analyzing more than a thousand records and establishing 2022 as the final year of analysis, with the aim of continuing to deepen the structure of knowledge of the study of the SDGs in the business field.

**Table 1.** Reviews indexed in the Web of Science core collection on the SDGs at the company level.

| Authors | Journal | Research Objective | Type of Review | Period Analyzed | Papers Analyzed |
|---|---|---|---|---|---|
| Di Vaio et al. [25] | Journal of Business Research | The research explores the relationship between artificial intelligence and rapid advances in machine learning to achieve sustainable resource management in line with the SDGs. | Bibliometric analysis | 1990–2019 | 73 |
| Pizzi et al. [28] | Journal of Cleaner Production | This review systematically examines, using bibliometric and systematic literature review methods, the scientific knowledge on the SDGs and the business sector. | Bibliometric and systematic review | 2012–2019 | 266 |
| Ye et al. [29] | Journal of Cleaner Production | The study reviews the literature on the link between CSR and sustainable development within the framework of the SDGs developed by the United Nations. | Bibliometric analysis | 1997–2019 | 1006 |
| Alvino et al. [30] | Journal of Intellectual Capital | The article analyses whether intellectual capital, through the application of knowledge management processes, can influence business orientation towards the creation of sustainable business models and, therefore, the achievement of the SDGs. | Systematic review | 1990–2019 | 45 |
| Di Vaio et al. [31] | Meditari Accountancy Research | The aim of this article is to provide a comprehensive and systematic overview of the academic literature focusing on the role of integrated reporting and integrated thinking in achieving sustainable business models for the SDGs. | Bibliometric analysis | 1990–2019 | 60 |

**Table 1.** *Cont.*

| Authors | Journal | Research Objective | Type of Review | Period Analyzed | Papers Analyzed |
|---|---|---|---|---|---|
| Meseguer-Sánch et al. [32] | Sustainability | This research aims to analyze the relationship between the concepts of CSR and CS in order to understand the advances in current scientific production as well as their link with the SDGs. | Bibliometric analysis | 2001–2020 | 3079 |
| Agrawal et al. [33] | Business Strategy and the Environment | The research aims to analyze the literature on the adoption and implementation of the SDGs in companies' supply chain activities. | Bibliometric analysis | 2015–2021 | 144 |
| Del Giudice et al. [34] | Maritime Policy and Management | This work aims to investigate through a barometric analysis whether digitalization and new technologies can help in the creation of sustainable business models and, therefore, the fulfilment of the SDGs. | Bibliometric analysis | 1969–2020 | 132 |
| Garrido-Ruso et al. [26] | Sustainability | This paper aims to determine the scope of the existing literature on the role of organizations in contributing to the advancement of the SDGs. | Bibliometric analysis | 2015–2021 | 543 |
| Jan et al. [35] | Environmental Science and Pollution Research | The research reviews the academic literature concerning the effect of CS on sustainable development and the achievement of the SDGs. | Bibliometric analysis | 2005–2021 | 1214 |
| Lee et al. [27] | Sustainability | The research aims to deepen through a systematic review the current state of sustainability and SDG studies in business and management disciplines. | Systematic review | 2015–2021 | 237 |
| Kurek et al. [37] | Sustainability | The aim of this article is to improve the understanding of how design thinking and its set of tools and methods contribute to the creation and innovation of sustainable business models and to the achievement of the SDGs. | Bibliometric and systematic review | 2002–2021 | 371 |

Source: own elaboration.

## 3. Methodology

The bibliometric analysis carried out in this research is based on the academic literature found in the WOS by applying Boolean (AND; OR), proximity (NEAR/5) and marker operators (*; $) [37]. This database has been selected as it is one of the most restrictive for the indexing of works. Thus, it is assumed that the potential results are of sufficient scientific quality. Once in WOS, the Web of Science Core Collection was used as the query database for five main reasons: (1) the wide range of scientific journals, (2) the systemic and dynamic method of journal selection, (3) the multitude of scientific disciplines, (4) the inclusion of a significant number of academic and research institutions, and (5) the availability of the citation network [38]. In particular, the following indexes from the main collection were used: Science Citation Index Expanded (SCI-E), Social Sciences Citation Index (SSCI), Emerging Sources Citation Index (ESCI), Conference Proceedings Citation Index-Science (CPCI-S), Book Citation Index (BCI) and Arts and Humanities Citation Index (AHCI).

After determining the use of the Web of Science Core Collection, we proceeded to search for those papers that were of interest for the subject analysis. After several tests in which both the most relevant results (according to the WOS algorithm) and the less significant ones (to discard possible unrelated results) were analyzed, it was determined that the best search equation among those considered was the following:

TS = (((sustainab* development goal$) NEAR/5 (compan* OR firm$ OR corporation$ OR business*)))

Analyzing the algorithm, it can be seen that the search was divided into two main groups: SDGs and companies. To avoid possible omissions, several terms were used, which were also checked one by one in the Thesaurus web dictionary of synonyms and antonyms to corroborate that all terms likely to provide results of interest had been considered. In addition, the NEAR operator with a value of 5 was used, which limited the valid papers to those that presented results from both groups separated by five or fewer words [38], thus avoiding unrelated topics that could casually present such terms. At the same time, the so-called wildcard (*) was used to include possible variations of words in the valid results [39]. Finally, it should be mentioned that, within the groups, the Boolean operator OR was used when dealing with synonyms. These parameters were applied to the topic, which includes the title and the abstract of the papers, as well as the keywords provided by the author and those included by WOS itself until 2022.

Following the application of the search algorithm on 19 January 2023, a total of 4150 documents were accepted. To carry out the scientific production, the "Preferred Reporting Items for Systematic Reviews and Meta-Analyses" (PRISMA statement) was used. Through this methodology, the search, classification and refinement of the results obtained after the search could be carried out, as can be seen in Figure 1. The PRISMA statement was used because it offers the possibility of increasing the reliability and reproducibility of the reviews, because of its exhaustiveness of the method and because it is widely used to carry out bibliometric studies [40–42]. After filtering the data according to the categories Environmental Sciences, Business, Management and Economics and checking for duplicates, the number of documents was reduced from 4150 to 2366 items.

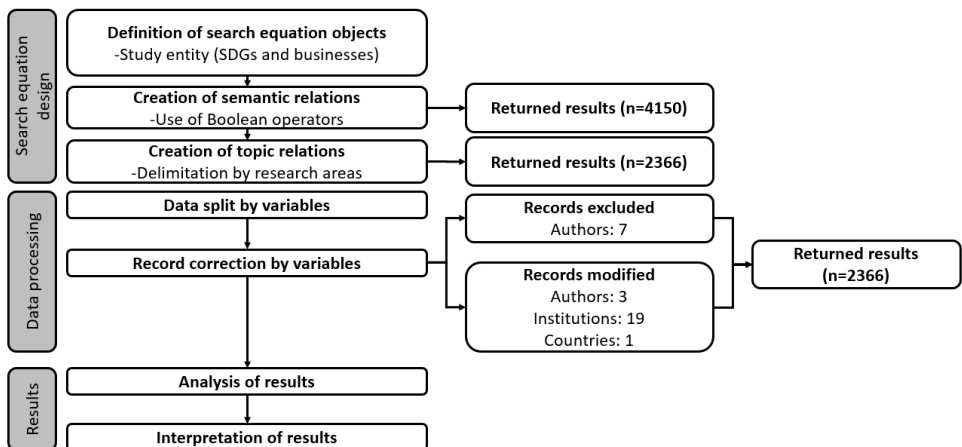

**Figure 1.** Flow diagram of the bibliometric review procedure developed. Source: own elaboration based on PRISMA guidelines.

To allow for an analytical reading, multiple classification variables were selected in order to detect both trends and magnitudes and, in this way, to understand the structure of knowledge on the scientific production that has analyzed the SDGs at the business level. Each of the variables used is detailed below. In order to find out what the interest in the subject had been over time, the records were segmented according to the year in which they were published in the different journals. In this way, it is possible to observe both the moment when the first results appear and the point at which the subject studied begins to arouse considerable interest, as well as the current state. In this case, the period covered ends in 2022, as this is the last year to present complete data. Another method used is the disaggregation according to the type of document. It should be borne in mind that a record may belong simultaneously to two or more groups, so that, when looking at the figures, the sum of the value of the groups is greater than the actual number of records due to this duplication.

In a similar way to the previous one, the different records were distinguished according to the area or areas of knowledge to which they belonged. In this sense, the WOS

classification system (known as WOS Categories) was used. The analysis was also complemented with a network map, carried out using the VOSviewer tool in its version 1.6.18. In particular, a network map of keyword co-occurrence was used, including the keywords that appeared at least five times in the records considered and identifying clusters based on the established parameters. VOSviewer is a software application for constructing and visualizing bibliometric networks [43]. These networks can include journals, researchers or individual publications, and can be constructed on the basis of citation, bibliographic linkage, co-citation or co-authorship relationships [44]. As with the document typology, duplicities are found in some records as they belong simultaneously to several of these areas. In addition, it should be noted that the WOS categories were used instead of the research areas offered by the database because the former had a higher level of disintegration once this filter was applied to the results obtained. Authorship, for its part, was one of the most important variables to be considered. The classification both by number of publications related to the subject of study and by the citations obtained by these works made it possible to discern who the leading figures in the field were, as well as the institutions (analyzed separately in another section) with which they were associated. In addition, by using a co-citation map, carried out through the VOSviewer application, the connections between the authors appearing in the bibliographical references of the records studied were analyzed. In particular, the mapping of the co-citation network for authors was carried out with a minimum of 30 citations, identifying existing clusters on the basis of this parameter. It is, therefore, another measure of influence, since it does not analyze the direct contributions to the scientific production analyzed, but rather the basis contributed to the works on this subject, so that authors may appear who do not deal with the object of study, but who indirectly contribute to its development.

Institutions (known in WOS as affiliations) are the various bodies to which the authors who carry out the research work belong. These are mainly universities, although others such as research centers may also appear. Their study made it possible to identify pioneering organizations in the study of the SDGs in the business sphere. In addition, they were a first indicator for the country ranking. In particular, the number of publications held by each institution was analyzed. It should be noted that some of the data obtained from the WOS were provided in cluster form as well as at sub-organizational (campus) levels. To avoid distortions in the analysis due to duplication, these groupings were removed and the records were reassigned to the organizations to which they actually belonged. Publication sources (publication titles or publication titles in WOS) are the media where the different papers are presented. These are mainly scientific journals, although others, such as conference proceedings, can also be found. In this sense, the main sources of publication on the subject under study were studied. Similarly, the volume of records was briefly analyzed according to the publisher to which these sources belonged. However, this variable was not explored further as it depends, to a large extent, on the number of sources owned by each publisher, which could distort interest by not considering the size of the latter. Finally, one of the variables of greatest interest was analyzed: the geographical classification by country and region. The study of the publishing strength of each territory made it possible to detect where the main advances in the field were to be found.

## 4. Results and Discussion

Scientific production linked to the SDGs at the business level began to intensify from 2015 onwards, rising from 73 articles in 2015 to 463 in 2022, a more than 6-fold increase in that period (see Graph 1). This is due to the fact that these goals were approved by the United Nations Summit on Sustainable Development, held from 25 to 27 September 2015 in New York. As a result of this summit, the United Nations General Assembly set out the goals in the famous document entitled "Transforming our world: the 2030 Agenda for Sustainable Development", better known as the "2030 Agenda" [45].

However, although the SDGs were formulated in 2015 by the United Nations, the historical series on the scientific production analyzed begins in 1992, as can be seen in

Figure 2. This is because in 1992, at the United Nations Conference on Environment and Development in Rio de Janeiro, the idea of developing an agenda, Agenda 21, which would address urgent problems and prepare the world for the 21st century, had already been discussed, as it had been recognized that humanity was facing a defining moment in its history [46]. Global challenges such as increasing inequalities between and within nations, worsening poverty, hunger, disease and the continuing deterioration of ecosystems on which human well-being depends required the attention of all nations [47]. The concern to integrate the environment and development into a prosperous future was embodied in Agenda 21, a document that for the first time reflected a political commitment and global consensus with a high level of cooperation for development and the environment. The agenda was organized into three sections: economic and social dimension, conservation and resource management for development and strengthening the role of major groups, totaling 31 programme areas, each with their respective objectives, activities and means of implementation [48].

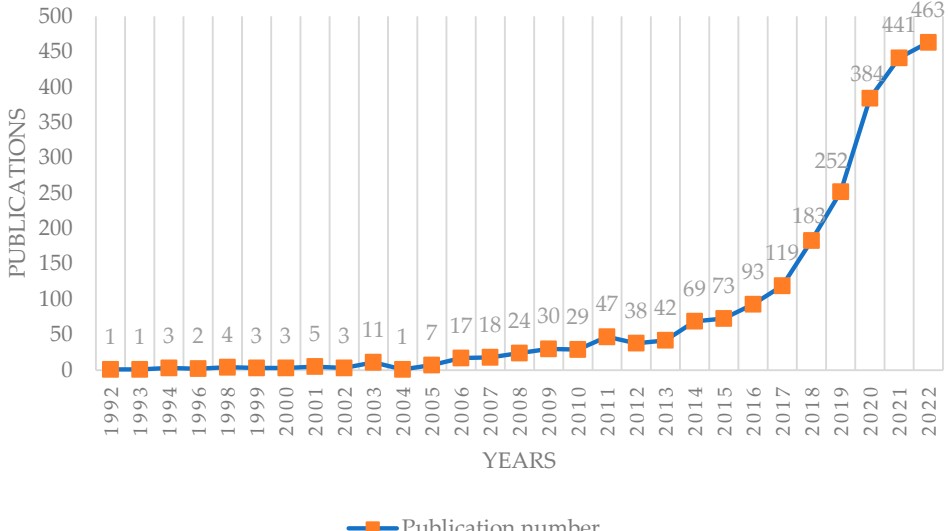

**Figure 2.** Scientific production analyzed by year of publication.

Figure 2 also shows a slight increase in the second half of the first decade of the 21st century that lasts until the first half of the second decade. This can be explained by two main reasons. On the one hand, in 2000, the MDGs, considered the predecessors of the SDGs, were established at the Millennium Summit [49]. These eight goals were the foundation of a global initiative to eradicate extreme poverty and hunger; achieve universal primary education and gender equality; reduce child mortality; improve maternal health; combat HIV/AIDS, malaria and other diseases; achieve environmental sustainability; and develop a global partnership for development [50]. Furthermore, in 2012, the SD21 Project of the United Nations Conference on Sustainable Development set out the coherent vision of Sustainable Development for the new century and established a way forward for the international community, national governments, businesses, partnerships and other stakeholders [51]. Likewise, it is worth highlighting the incessant growth in the number of journals indexed in the main collection of the WOS in the last decade [37], which has led to an increase in the scientific production present in the database, meaning an increase in the scientific production analyzed in this study.

Regarding the distribution of scientific output by publication format used, as can be seen in Figure 3, journal articles are the main source of dissemination of research on the contribution of business to the SDGs, with 1872 results (accounting for 79.12% of the 2366 results returned by the search equation). Proceedings articles and article reviews also stand out, accounting for 358 (15.13%) and 160 (6.76%) of the records, respectively, demonstrating that the structure of the publication format does not differ from that of

other research fields. The rest of the results are divided between early access articles, book chapters, editorial materials, books, book reviewers and corrections.

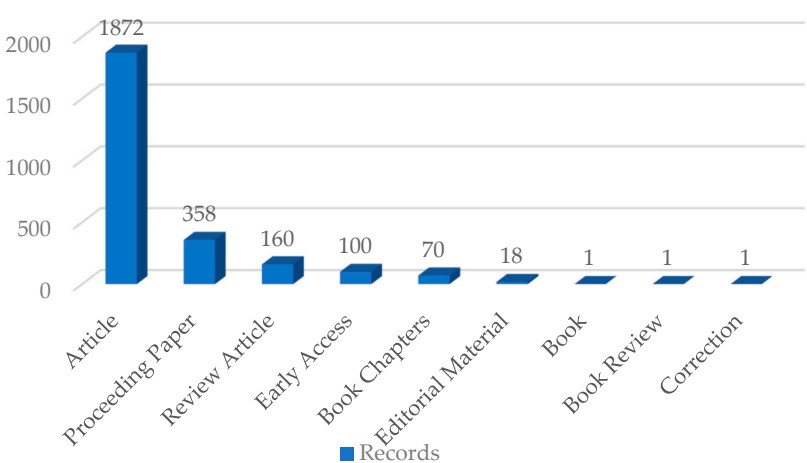

**Figure 3.** Number of records by publishing format used.

Table 2 shows the distribution of the work by research field. As might be expected, among the top ten fields are the fields used for filtering in the search equation. However, among the top ten there are other categories (and therefore potential synergies) with the fields used for the development of the literature review. In this sense, Environmental Science is the most recurrent area (1042 results), followed by Green Sustainable Science Technology (853 results) and Business (751 results). Interest in the field of Management (715) and Economics (409) is also noteworthy, as companies can adopt the SDGs as a framework to improve their economic, social and environmental performance (micro approach) and governments can use the SDGs as a mechanism to foster economic progress in different sectors of the economy (macro approach).

**Table 2.** Number of results by research fields (Top 30).

|  | WOS Categories | Records |  | WOS Categories | Records |
|---|---|---|---|---|---|
| 1 | Environmental Sciences | 1042 | 16 | Ecology | 21 |
| 2 | Green Sustainable Science Technology | 853 | 17 | Computer Science Interdisciplinary Applications | 19 |
| 3 | Business | 751 | 18 | Hospitality Leisure Sport Tourism | 18 |
| 4 | Environmental Studies | 721 | 19 | Energy Fuels | 17 |
| 5 | Management | 715 | 20 | Public Environmental Occupational Health | 17 |
| 6 | Economics | 409 | 21 | Computer Science Information Systems | 15 |
| 7 | Engineering Environmental | 282 | 22 | Social Sciences Interdisciplinary | 15 |
| 8 | Business Finance | 66 | 23 | Psychology Applied | 14 |
| 9 | Regional Urban Planning | 53 | 24 | Water Resources | 14 |
| 10 | Ethics | 48 | 25 | International Relations | 13 |
| 11 | Operations Research Management Science | 32 | 26 | Geography | 11 |
| 12 | Education Educational Research | 31 | 27 | Agricultural Economics Policy | 10 |
| 13 | Engineering Industrial | 31 | 28 | Transportation | 10 |
| 14 | Development Studies | 29 | 29 | Forestry | 9 |
| 15 | Information Science Library Science | 26 | 30 | Law | 9 |

The results relating to the research fields highlight the multidisciplinary nature of the study of the SDGs at the business level, given that, among other aspects, these objectives may involve improving the conditions of workers (Ethics, Environmental Occupational Public Health and Applied Psychology), improving the environment in which the organization operates, as well as making better use of the resources employed by the organization (Energy Fuels, Water Resources and Ecology) or improving business productivity and profitability (Operations Research, Management Science, Industrial Engineering and International Relations). However, some of these research fields show a high potential to broaden the knowledge of the research topic addressed in this paper. For example, those areas that focus on the use and development of new technologies can contribute by analyzing ways to improve and optimize companies' compliance with the SDGs. Studies on development and urbanism may also be of interest for future research focusing on the effect of business agglomeration on SDG compliance.

In order to graphically illustrate the connection between the study of the SDGs at the business level and other topics, a keyword co-occurrence analysis was carried out. In this way, it was possible to identify the relationships between keywords of the topic under study, based on the proximity within the map. As can be seen from Figure 4, there is a strong connection between the keywords sustainability and sustainable development, as these concepts are two ways of understanding the purpose of the SDGs. Furthermore, these keywords are linked, among others, with the words CSR and CS (representing complementary frameworks for achieving the SDGs), digital economy and digitalization (representing two means to enhance the achievement of the SDGs at the business level) or performance and competitive advantage (being two benefits that adherence to the SDGs by companies can provide).

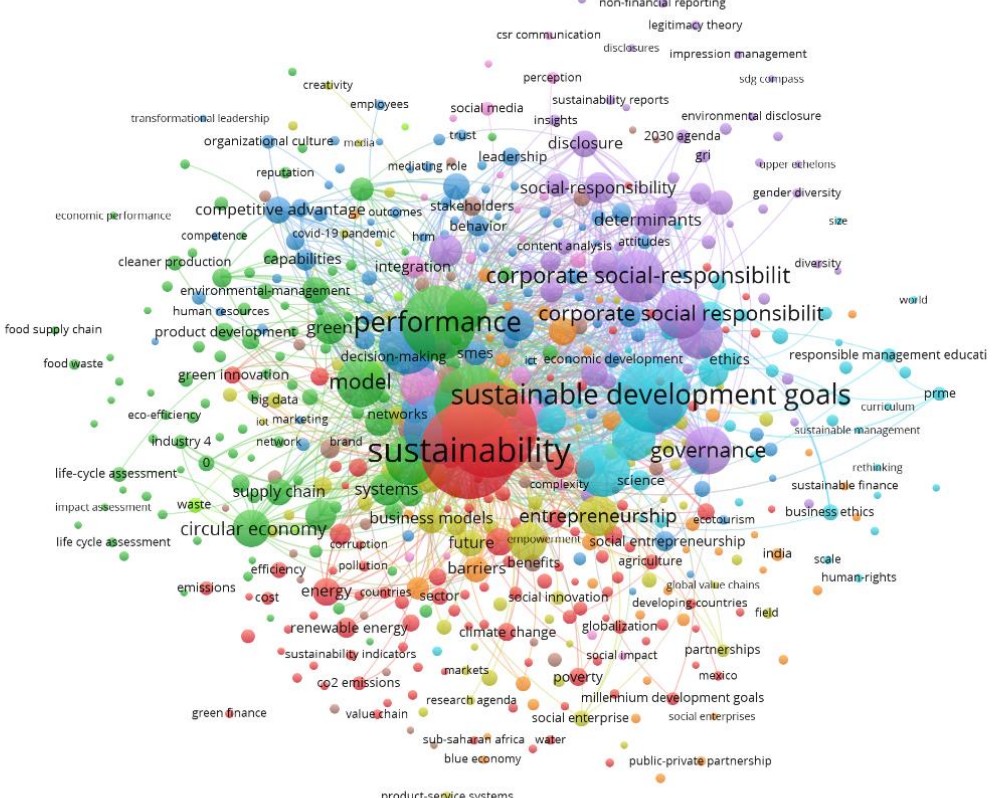

**Figure 4.** Network map of the co-occurrence of keywords. For practical reasons, we have included those keywords that appear at least 5 times in the records considered (unit of analysis: all keywords). The size of the nodes is proportional to the number of times the keyword appears.

Likewise, the analysis of keywords is complemented with the study of their frequency, their belonging to the clusters as well as the analysis of their use over time. On the one hand, Table 3 shows the 20 most used keywords in the scientific production analyzed, as well as their belonging to each of the 10 clusters. The most used keywords for each cluster are: sustainability (Cluster 1), performance (Cluster 2), innovation (Cluster 3), entrepreneurship (Cluster 4), corporate social-responsibility (Cluster 5), sustainable development goals (Cluster 6), barriers (Cluster 7), benefits (Cluster 8), integration (Cluster 9) and social impact (Cluster 10). As can be seen in Table 3, the 20 most used keywords fall within the first 6 clusters, with no keywords belonging to the remaining 20 clusters. Similarly, it should be noted that Cluster 2 is the one with the highest number of keywords in the Top 20. On the other hand, Figure 5 shows the keyword overlap analysis in order to examine keyword usage over time. As can be seen, while the more classic keywords would be related to competitive advantage and organizational culture, the newer ones would be related to sustainable development objectives, the latter being more topical than sustainability and CSR as it represents a new framework established for understanding sustainability [8]. Likewise, competitive advantage and organizational culture have been widely discussed in the field of strategic management for more than four decades [5], which justifies that the key words for these concepts are less current.

**Table 3.** Frequency and cluster analysis according to keywords (Top 20).

| Keyword | Frequency | Cluster |
| --- | --- | --- |
| Sustainability | 426 | 1 |
| Sustainable development | 402 | 1 |
| Sustainable Development Goals | 268 | 6 |
| Management | 264 | 1 |
| Performance | 263 | 2 |
| Innovation | 228 | 3 |
| Impact | 163 | 3 |
| Business | 146 | 6 |
| Corporate Social Responsibility | 140 | 5 |
| Framework | 133 | 3 |
| Model | 127 | 2 |
| Governance | 119 | 6 |
| CSR | 111 | 5 |
| SDGs | 102 | 6 |
| Entrepreneurship | 91 | 4 |
| Corporate Sustainability | 90 | 5 |
| Strategy | 90 | 2 |
| Challenges | 83 | 4 |
| Circular economy | 78 | 2 |
| Implementation | 50 | 2 |

With regards to the main authors researching the subject analyzed, the results obtained in Table 4 show that the main author is Assunta Di Vaio with 10 entries, followed by Roberta Costa (8 entries), Armando Calabrese, Armando Calabrese, Rohail Hassan and Walter Leal (the latter four authors with 7 entries). As can be seen, the difference in entries between the top ten authors is small, varying by only four publications. The three authors with the most citations (excluding self-citations) are Sachin Kumar Mangla with 334, followed by Walter Leal (312 citations) and Osvaldo Luiz Goncalves Quelhas (302 citations). As for the

country of the institutions to which the 30 main authors on the subject belong, it is worth noting that nine researchers are affiliated with Italian organizations and two to German, British, Spanish and Chinese institutions.

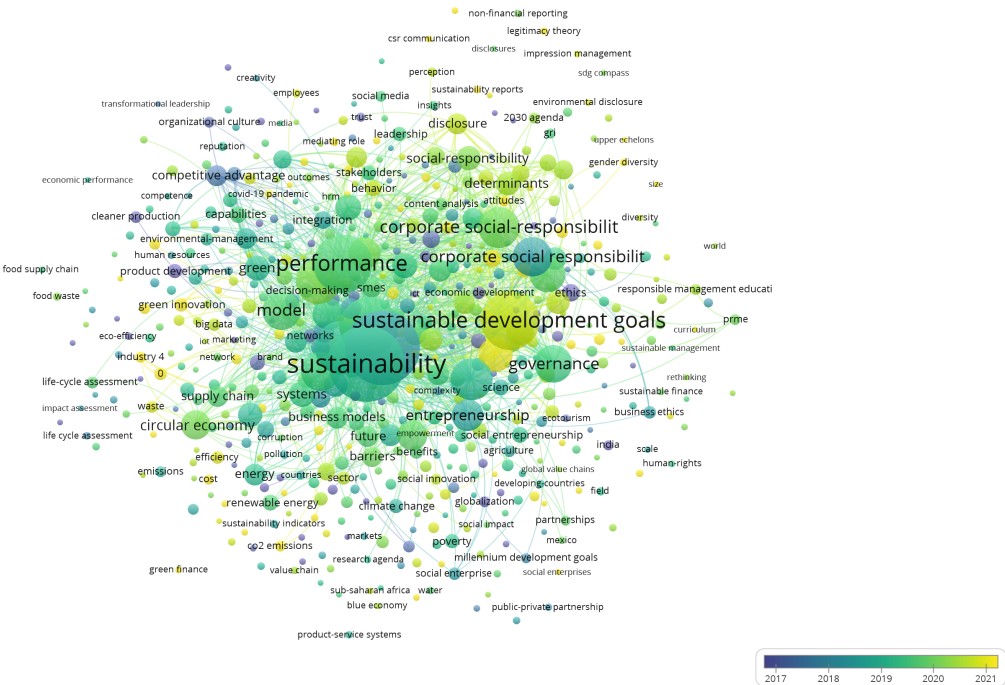

**Figure 5.** Overlay map of the co-occurrence of keywords. For practical reasons, we have included those keywords that appear at least 5 times in the records considered (unit of analysis: all keywords). The size of the nodes is proportional to the number of times the keyword appears.

Although it is necessary to expose the content included in Table 4, the interest of the authors' analysis remains in the co-citation relationships shown in Figure 6. The co-citation analysis developed allows us to discover the number of times two authors are co-cited by other scholars and, consequently, and is also used as a proxy to establish possible relationships in their lines of research. According to the number of co-citations, Michael Porter, Stefan Schaltegger, Archie Carroll, the United Nations and the European Commission appear as the most influential in the research area analyzed as they have the highest number of co-citations with the rest of the main authors, thus differing from the pattern shown in Table 4. This is due to the strong influence of the three authors and the two institutions in the field of SDGs and competitive advantage for organizations. Thus, the United Nations is the author of the founding texts of the SDGs and the Brundland Report in which the foundations of sustainability and sustainable development are laid [4,5]; the European Commission is the author of the main reports to achieve sustainability and sustainable development within the European Union SDGs [52–54]; Michael Porter is considered the father of business strategy given his research to investigate the achievement of business competitive advantages [54]; Stefan Schaltegger is the author of the main research on CS [55]; and Archie Carroll is one of the great scholars of CSR [56]. This makes the named authors and institutions particularly relevant to the topic under discussion, and they are therefore cited together when developing research on the subject, since their citation allows us to draw on the foundational texts of the SDGs, sustainability and sustainable development, while explaining their relevance for gaining competitive advantage, as well as their possible integration within CSR and CS programmes. Moreover, it should also be noted that the five authors share the importance of improving the competitive position of organizations through respect for profanity and respect for the environment in which companies operate, showing that the principles that govern sustainability can be a means

of business differentiation, as well as a way to guarantee the sustainable competitive advantage of companies over time.

**Table 4.** Number of records, citations, work impact and affiliation country for each main author (Top 30).

| Author | Records | Citations * | Ratio | Institution | Country |
|---|---|---|---|---|---|
| Assunta Di Vaio | 10 | 227 | 22.7 | Parthenope University Naples | Italy |
| Roberta Costa | 8 | 184 | 23.0 | University of Rome Tor Vergata | Italy |
| Armando Calabrese | 7 | 202 | 28.9 | University of Rome Tor Vergata | Italy |
| Rohail Hassan | 7 | 217 | 31.0 | Universiti Utara Malaysia | Malaysia |
| Walter Leal | 7 | 312 | 44.6 | Manchester Metropolitan University | United Kingdom |
| Marga Hoek | 6 | 6 | 1.0 | Johannes Gutenberg University of Mainz | Germany |
| Liu Y | 6 | 126 | 21.0 | Linkoping University | Sweden |
| Sachin Kumar Mangla | 6 | 334 | 55.7 | Jindal Global University | India |
| Osvaldo Luiz Goncalves Quelhas | 6 | 302 | 50.3 | Universidade Federal Fluminense | Brazil |
| Rob Van Tulder | 6 | 238 | 39.7 | Erasmus University Rotterdam | The Netherlands |
| Anthony Alexander | 5 | 96 | 19.2 | University of Sussex | United Kingdom |
| Rosley Anholon | 5 | 109 | 21.8 | Universidade Estadual de Campinas | Brazil |
| Alexander Brem | 5 | 117 | 23.4 | University of Stuttgart | Germany |
| Thomas Dyllick | 5 | 158 | 31.6 | University of St Gallen | Switzerland |
| Isabel Maria Garcia-Sánchez | 5 | 122 | 24.4 | University of Salamanca | Spain |
| Nathan Levialdi Ghiron | 5 | 58 | 11.6 | University of Rome Tor Vergata | Italy |
| Lea Iaia | 5 | 14 | 2.8 | University of Chieti-Pescara | Italy |
| Ivan Montiel | 5 | 57 | 11.4 | Loyola Marymount University | United States |
| Rosa Palladino | 5 | 216 | 43.2 | University of Milano-Bicocca | Italy |
| Izabela Simon Rampasso | 5 | 109 | 21.8 | Universidad Católica del Norte | Chile |
| Gilberto Santos | 5 | 21 | 4.2 | Polytechnic Institute Cavado Ave | Portugal |
| Demetris Vrontis | 5 | 50 | 10.0 | University of Nicosia | Cyprus |
| Zhu L | 5 | 95 | 19.0 | Wuhan Textile University | China |
| Jaffar Abbas | 4 | 13 | 3.3 | Shanghai Jiao Tong University | China |
| Beatriz Aibar-Guzmán | 4 | 104 | 26.0 | University of Santiago De Compostela | Spain |
| Andrew Alola | 4 | 68 | 17.0 | University of Vaasa | Finland |
| Stefano Amelio | 4 | 8 | 2.0 | University of Insubria | Italy |
| Surajit Bag | 4 | 64 | 16.0 | University of Johannesburg | South Africa |
| Francesco De Luca | 4 | 15 | 3.8 | University of Chieti-Pescara | Italy |
| Patrizia Gazzola | 4 | 110 | 27.5 | University of Insubria | Italy |

* Self-citations have been excluded.

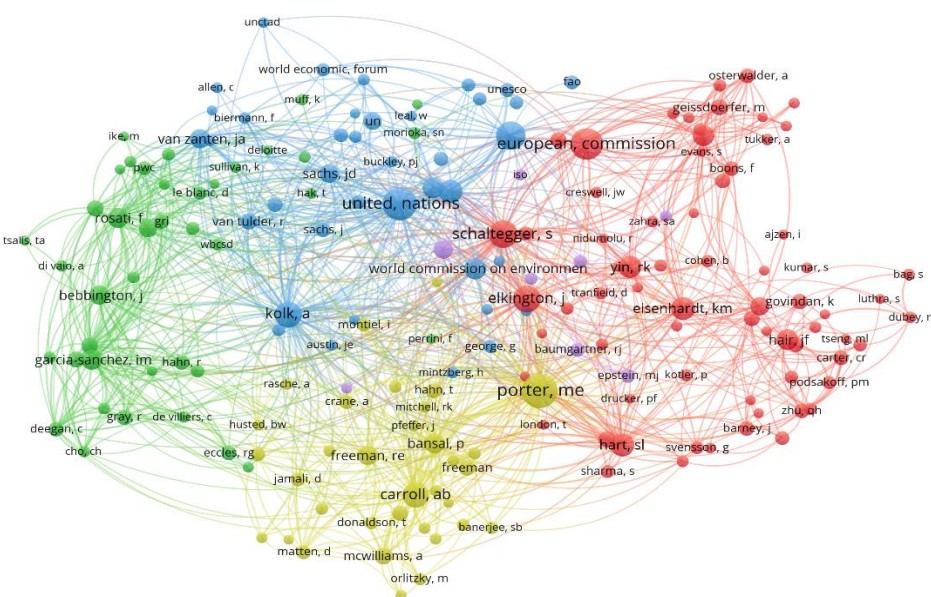

**Figure 6.** Co-citation network map for authors with a minimum of 30 citations.

With regards to the main institutions that support SDG research at the business level, as can be seen in Table 5, the leading institution in terms of the number of publications is the Bucharest University of Economic Studies (44 publications), followed by the Indian Institutes of Technology (38 publications) and the Ministry of Education Science of Ukraine (34 publications). As can be seen, there is only one country that has two institutions among the ten with the highest scientific production on the subject analyzed, and that is Italy (Parthenope University Naples and the University of Rome Tor Vergata). This is in line with Table 4, given that the transalpine country has the largest number of academics on the subject analyzed. It is also important to highlight the role played by the United Kingdom, since it has 4 of the 30 institutions with the greatest specialization in the subject (University of London, University College London, University of Oxford and University of Oxford).

With regards to the top journals for disseminating research results, Table 6 shows that Sustainability (504 publications) has the highest number of contributions, followed by Journal of Cleaner Production (217) and Business Strategy and the Environment (59). Of the top 30 journals, 25 have a Journal Impact Factor (JIF), a measure applied to those journals considered to have sufficient quality and research impact (Clarivate, 2021). Specifically, 14 belong to the 1st quartile in at least one of their categories, while 9 belong to the 2nd quartile, which means that they are in the top 25% and 50% of the journals previously considered to have sufficient quality and impact, being mostly journals specializing in environmental research and, to a lesser extent, in the study of business. With regard to the weight of the importance of publishers by academic publications, as can be seen in Figure 7, it is worth noting that MDPI occupies the first position, with 537 publications, followed by Elsevier (437), Emerald (245), Springer (206) and Wiley (157).

In terms of the geographical distribution of the scientific production analyzed, Figure 8 shows that the United States is the country with the greatest contribution in terms of works. A total of 259 publications are related to American institutions, followed by the United Kingdom with 247 and China with 236. In fact, these three countries account for 31.36% of all existing scientific production on the subject. The role of Italy (198), Spain (178) and Germany (119) also stands out, both in terms of scientific production, occupying fourth, fifth and sixth place, respectively, and in terms of the number of relevant authors and institutions focused on the subject under study.

**Table 5.** Institutions by number of records and region (Top 30).

| Institutions | Records | Region |
|---|---|---|
| Bucharest University of Economic Studies | 44 | Romania |
| Indian Institutes of Technology | 38 | India |
| Ministry of Education Science of Ukraine | 34 | Ukraine |
| Universidade de Sao Paulo | 24 | Brazil |
| University of London | 21 | United Kingdom |
| Parthenope University Naples | 19 | Italy |
| University of Rome Tor Vergata | 17 | Italy |
| University of Southern Denmark | 17 | Denmark |
| Erasmus University Rotterdam | 15 | The Netherlands |
| Technical University Czestochowa | 15 | Poland |
| Tecnologico de Monterrey | 14 | Mexico |
| University of Johannesburg | 14 | South Africa |
| Monash University | 13 | Australia |
| University Vollege London | 13 | United Kingdom |
| University of Oxford | 13 | United Kingdom |
| University of Zilina | 13 | Slovakia |
| HSE University | 12 | Russia |
| Universidade de Lisboa | 12 | Portugal |
| Universiti Utara Malaysia | 12 | Malaysia |
| University of Sussex | 12 | United Kingdom |
| Warsaw School of Economics | 12 | Poland |
| Copenhagen Business School | 11 | Denmark |
| Norwegian University of Science Technology | 11 | Norway |
| Peter the Great St.Petersburg Polytechnic University | 11 | Russia |
| University of Erlangen Nuremberg | 11 | Germany |
| University of Granada | 11 | Spain |
| Josip Juraj Strossmayer University of Osijek | 11 | Croatia |
| University of Queensland | 11 | Australia |

**Table 6.** List of journals by number of records (Top 30) and their 2021 Journal Impact Factor (JIF) quartile.

| Journals | Records | Highest 2021 JIF Quartile | Publishing Houses |
|---|---|---|---|
| Sustainability | 504 | Q2 | MDPI |
| Journal of Cleaner Production | 217 | Q1 | Elsevier |
| Business Strategy and the Environment | 59 | Q1 | Wiley |
| Corporate Social Responsibility and Environmental Management | 37 | Q1 | Wiley |
| Journal of Business Ethics | 34 | Q1 | Springer |
| International Journal of Management Education | 22 | Q1 | Elsevier |
| Environment Development and Sustainability | 19 | Q2 | Springer |
| European Journal of Sustainable Development | 19 | n/d | European Center Sustainable Development |
| Technological Forecasting and Social Change | 18 | Q1 | Elsevier |
| Journal of Environmental Management | 17 | Q1 | Elsevier |
| Sustainability Accounting Management and Policy Journal | 16 | Q2 | Emerald |
| Environmental Science and Pollution Research | 15 | Q2 | Springer |
| International Journal of Environmental Research and Public Health | 15 | Q2 | MDPI |
| Business Strategy and Development | 14 | Q3 | Wiley |
| Corporate Governance the International Journal of Business in Society | 14 | n/d | Emerald |
| Resources Conservation and Recycling | 14 | Q1 | Elsevier |
| Economic Research Ekonomska Istrazivanja | 13 | Q2 | Routledge |
| Journal of Business Research | 13 | Q1 | Elsevier |
| Frontiers in Environmental Science | 12 | Q2 | Frontiers |
| Sustainability Science | 12 | Q1 | Springer |
| Amfiteatru Economic | 11 | Q2 | Editura |
| International Journal of Life Cycle Assessment | 11 | Q2 | Springer |
| Social Responsibility Journal | 11 | n/d | Emerald |
| Baltic Journal of Economic Studies | 10 | n/d | Routledge |
| Ecological Economics | 10 | Q1 | Elsevier |
| Energy Policy | 10 | Q1 | Elsevier |
| Journal of International Business Policy | 10 | Q1 | Springer |
| Economies | 9 | n/d | MDPI |
| Forest Policy And Economics | 9 | Q1 | Elsevier |

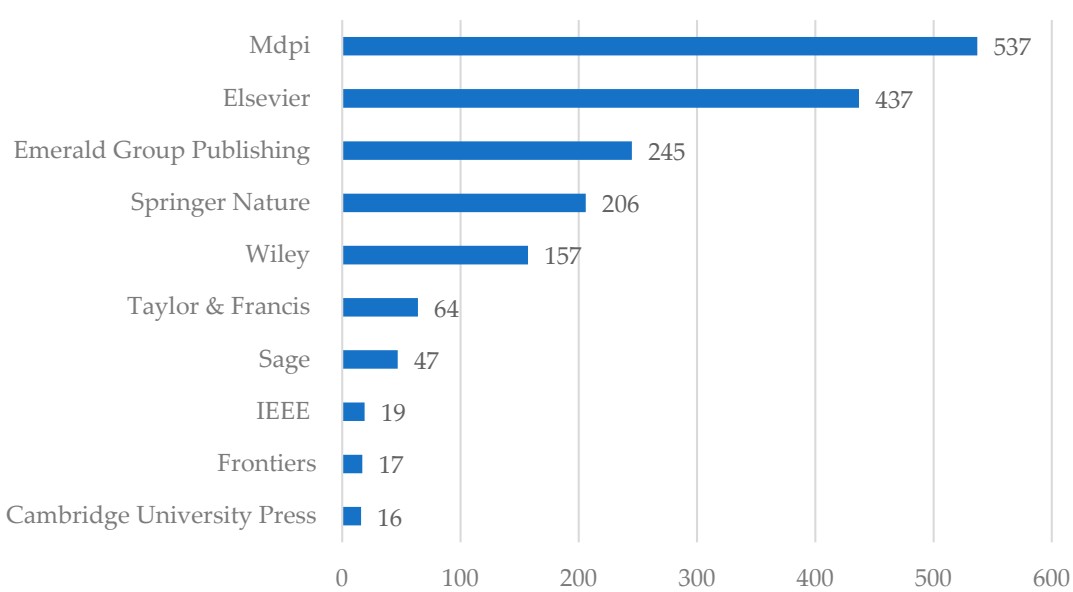

**Figure 7.** Top publishers by number of publications.

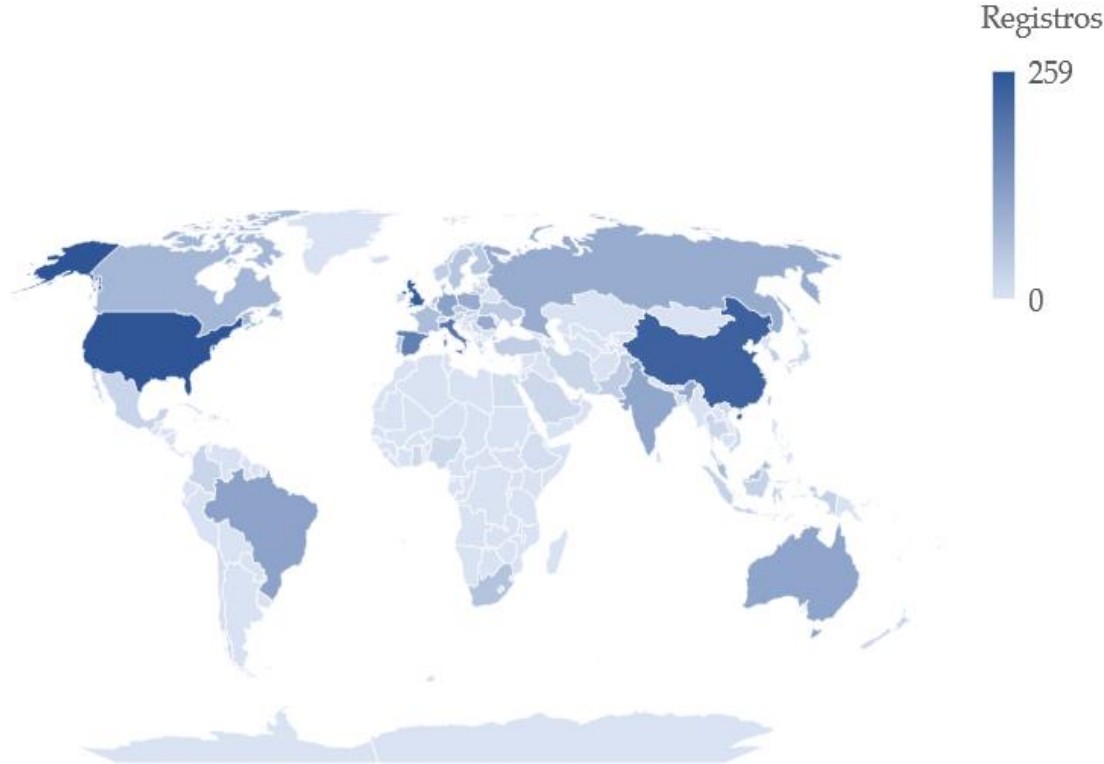

**Figure 8.** Number of records by countries.

## 5. Conclusions

This research allows for analyzing the structure of knowledge on the SDGs at the business level, making it useful both for neophyte academics who are beginning to address this line of research, and for experienced academics who wish to learn about the evolution of scientific production on the subject under analysis.

Studying the scientific literature that addresses the SDGs in the business context is important because it provides a deeper insight into how this topic is currently being addressed. This literature provides a wide range of information on the role and relevance

of the SDGs for organizations, helping to better understand how these initiatives are implemented in practice. It can help companies to improve their sustainability strategy, as it allows them to learn about best practices and lessons learned from other companies that have used the SDGs as a framework for achieving sustainability. Reviewing the scientific literature related to the SDGs can also help companies to identify opportunities for innovation and improvements, as well as to make the best use of existing resources. This enables organizations to improve their financial performance as well as linking their corporate image to sustainability. Furthermore, the study of this academic literature can help companies to better understand how the SDGs can contribute to the achievement of business goals, while boosting their long-term sustainability.

The research shows the existence of a process of intensification of scientific production since 2015, when the MDGs were replaced by the SDGs. In fact, from 2015 to 2022, there has been an increase in the scientific production analyzed of 534.25%, highlighting the exponential growth of the subject matter analyzed. In terms of the distribution of scientific output by publication format used, journal articles are the main source of dissemination of research on the contribution of business to the SDGs, followed by proceedings articles and article reviews. In terms of knowledge areas, Environmental Science is the most recurrent area, followed by Sustainable Green Technology Science and Business. Interest in the field of Management and Economics also stands out, as companies can adopt the SDGs as a framework to improve their economic, social and environmental performance (micro approach) and governments can use the SDGs as a mechanism to foster economic progress in different sectors of the economy (macro approach). Furthermore, these keywords are linked, among others, to the words CSR and CS (which represent complementary frameworks for achieving the SDGs), digital economy and digitalization (which represent two means to improve the achievement of the SDGs at the corporate level) or performance and competitive advantage (which are two benefits that companies' adherence to the SDGs can bring).

The main authors researching the analyzed topic are Assunta Di Vaio, followed by Roberta Costa, Armando Calabrese, Armando Calabrese, Rohail Hassan and Walter Leal. However, the difference in scientific production between the top ten authors is small, varying by only four publications. The co-citation analysis also shows that Michael Porter, Stefan Schaltegger, Archie Carroll, the United Nations and the European Commission are the most influential authors in the research area analyzed as they have the highest number of co-citations with the rest of the main authors. This is due to the strong influence of the three authors and the two institutions in the field of SDGs (United Nations and European Commission), CSR (Archie Carroll), CS (Stefan Schaltegger) and Corporate Competitive Advantage (Michael Porter). As for the main institutions supporting SDG research at the corporate level, the leading institution in terms of number of publications is the University of Economic Studies in Bucharest, followed by the Indian Institutes of Technology and the Ministry of Educational Sciences in Ukraine, with only one country having two institutions in the top ten with the highest scientific output on the topic analyzed: Italy (Parthenope University of Naples and the University of Rome Tor Vergata). The main journals for the dissemination of research results on the subject are Sustainability (MDPI), followed by Journal of Cleaner Production (Elsevier) and Business Strategy and the Environment (Wiley). In terms of the weight of importance of publishers by academic publications, MDPI ranks first, followed by Elsevier, Emerald, Springer and Wiley. Likewise, in terms of the geographical distribution of the scientific production analyzed, the United States is the country with the highest contribution in terms of number of papers, followed by the United Kingdom and China.

Future lines of research on the literature reviewed include studying the applicability of the SDGs across a wide range of sectors, the effects of SDG implementation on company performance, both financial and non-financial, as well as the impact of corporate governance on the SDGs and how companies can take advantage of these initiatives. There are also a number of research opportunities to examine best practices in SDG implementation

that have been developed in different sectors and countries. This could help companies to take better advantage of the sustainability opportunities offered by the SDGs, as well as maximize the benefits they derive from their adherence. Similarly, there are research opportunities to explore how companies can better communicate their sustainability efforts related to the SDGs, including how they can address the emerging challenges related to these initiatives.

This research presents several theoretical and practical contributions. First, the study advances the understanding of the scientific production that has focused its efforts on the study of the SDGs at the business level. Second, the research can be used by researchers to identify the main institutions and geographical regions for research stays and/or joint projects on the analyzed topic. Second, the study allows researchers to network with other researchers in case they want to conduct joint research and even organize conferences on the topic under study. Third, through this study, academics can identify which are the main journals and publishers to disseminate their research results. Fourth, the study allows the authors to learn about the main theoretical references when conducting their research. Fifth, the study allows the authors to learn about the multidisciplinary and gaps in the subject matter examined, and to identify new research gaps that will enable them to generate new knowledge on the study of the SDGs at the business level.

Despite the important contributions of this bibliographic analysis, it should be noted that the study also suffers from certain limitations. In this sense, it is worth pointing out the limitation inherent to literature reviews, given that the scientific production is analyzed quantitatively but the content of the articles is not analyzed in depth. In order to overcome this limitation, a systematic review of the literature is proposed as a future line of research in order to be able to identify the objectives, methodologies used and conclusions of the scientific production analyzed in this study.

**Author Contributions:** Conceptualization, J.M.-F. and B.M.-L.; methodology, E.S.-G.; software, L.A.M.-T.; validation, B.M.-L., E.S.-G. and J.M.-F.; formal analysis, L.A.M.-T.; investigation, L.A.M.-T.; resources, E.S.-G.; data curation, J.M.-F.; writing—original draft preparation, L.A.M.-T.; writing—review and editing, B.M.-L.; visualization, E.S.-G.; supervision, B.M.-L.; project administration, J.M.-F. All authors have read and agreed to the published version of the manuscript.

**Funding:** This research received no external funding.

**Institutional Review Board Statement:** The present study did not involve humans or animals.

**Informed Consent Statement:** Not applicable.

**Data Availability Statement:** The datasets used and analyzed during the current study are avail-able from the corresponding author on reasonable request.

**Conflicts of Interest:** The authors declare no conflict of interest.

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
