# Peer review of "Sustainable Development Goals in the Business Sphere: A Bibliometric Review"

_sustainability, doi:10.3390/su15065075_

Round 1

Reviewer 1 Report

This submission is rather descriptive, but that said it can be published in my opinion, with minor revisions.

The argument why this article (review) is necessary is rather weak.  The authors themselves mention 5 publications published in 2021, among which one based on more than 1000 publications. Why then is a new review necessary, covering the year 2022?

line 227. Just write a more than 6-fold increase  (534.25% is overly detailed).

line 233 the SDGs are not founded, but maybe formulated

line 234 How can something that happened in 1999 explain something that happened in 1992?

A general remark about growth: is the growth of the database taken into account?

The authors did not use the Book Citation Index nor the A&H Citation index. Yet in graph 2, they mention a book and in Table 2 law is one of the categories. 

Table 3.  I think "registers" is a bad translation from the Spanish.; see also Fig.2

Author Response

"The argument why this article (review) is necessary is rather weak.  The authors themselves mention 5 publications published in 2021, among which one based on more than 1000 publications. Why then is a new review necessary, covering the year 2022?”

Thank you very much for your comments, based on your suggestions before the introduction of the table we have introduced in the revised version the motivations justifying the need to carry out the research.

“line 227. Just write a more than 6-fold increase  (534.25% is overly detailed).”

Thank you very much for your comment. Based on your suggestion we have introduced the increase in the way that has been recommended.

“line 233 the SDGs are not founded, but maybe formulated”

Thank you very much for your comment. Based on your recommendation, we have changed this verb in the new revised version.

“line 234 How can something that happened in 1999 explain something that happened in 1992?”

Thank you very much for your comment. It is a mistake, the year we were referring to is 1992 (the year of the Rio de Janeiro summit). We have made the change in the revised version.

“A general remark about growth: is the growth of the database taken into account?”                

Thank you very much for your comment. Based on your recommendation we have taken into account the growth of the WOS database to argue for the growth of the scientific output analysed.

“The authors did not use the Book Citation Index nor the A&H Citation index. Yet in graph 2, they mention a book and in Table 2 law is one of the categories.” 

Thank you very much for your comment, it has certainly helped us to improve the clarity and quality of the research. We did take into account the Book Citation Index and the A&H Citation index, however we did not introduce it in the methodological section, it was a mistake. Therefore, in the revised version we have introduced the use of these indexes for the search. Thank you again for your comment.

“Table 3.  I think "registers" is a bad translation from the Spanish.; see also Fig.2”

 Thank you very much for your comment, indeed the correct word is not "registers", but records. We have made the changes in the revised version.

Thank you very much for your efforts in reviewing the research.

Reviewer 2 Report

Sustainable Development Goals in the Business Sphere: A Bibliometric Review

by Javier Martínez-Falcó et al.

Overview

The manuscript concerns the analysis of the scientific landscape regarding the SGDs in the business area. To perform the study the authors made use of the bibliometric approach.

I find the manuscript quite well-written.  The aims of the work are clear, the Methodology is sound although it needs some clarifications, the Results need to be improved as well as the Conclusions.

Detailed comments

Abstract.

The authors should cite the database used (WoS) to perform the bibliometric analysis.

3. Methodology.

Line 128-129. The statement needs to be explained better. Furthermore, a bibliographic reference is required.

Line 156. 19 October 2022?

Regarding VOSViewer, the authors must indicate all the software parameters used (e.g. cluster resolution, minimum number of items for clusters). Specify whether default parameters were used. Please also indicate the software version used.

Results and discussionsThere is a confusion with the numbering of the figures. Also, sometimes the word "graph" is used, in other cases the authors use "Figure": please refer to the term "Figure". In addition, figure captions should be more comprehensive.

Figure 2 (pag. 10). Which keywords were used? Author keywords? Also, the explanation of the map is poor. I suggest adding a table with a list of the main keywords, their frequency and the cluster they belong to. In the main text, it is also  necessary to link the discussion of Figure 2  to the literature.

It would be interesting to add the 'overlay map' built by using VOSViewer to identify the "historical" keywords and the keywords emerging in the recent years. This map will be discussed.

Figure 1 (pag.12). Which research interests do the authors belonging to the four main clusters share?

5. Conclusions

The conclusions are just a summary of the findings. It is necessary to add some personal deductions and comments which also concern suggestions for the scientific community, policymakers and stakeholders. For example, which could be the future directions of the research? Which research lines should be strengthened?

Author Response

“Abstract.: The authors should cite the database used (WoS) to perform the bibliometric analysis.”

Thank you very much for your suggestion, based on your recommendation, we have introduced the database used in the abstract.

“3. Methodology.: Line 128-129. The statement needs to be explained better. Furthermore, a bibliographic reference is required.”

Thank you very much for your recommendation, based on your suggestion we have clarified the Boolean, proximity and marker operators used. In addition, we have introduced a reference explaining how these operators and markers work. Your comment has certainly helped us to improve the clarity of the study.

“Line 156. 19 October 2022?”

Thank you very much for detecting this error, the search was carried out on 19 January, we have modified it in the revised version. Thank you again.

“Regarding VOSViewer, the authors must indicate all the software parameters used (e.g. cluster resolution, minimum number of items for clusters). Specify whether default parameters were used. Please also indicate the software version used.”

Thank you very much for your recommendation, it has indeed contributed to improving the quality of the research. Based on your suggestions, we have introduced the VosViewer version as well as the parameters that were used to make the bibliographic maps.

Results and discussions. There is a confusion with the numbering of the figures. Also, sometimes the word "graph" is used, in other cases the authors use "Figure": please refer to the term "Figure". In addition, figure captions should be more comprehensive.”

Thank you very much for your comment, in the modified version we have changed all "graphics" to "figures".

“Figure 2 (pag. 10). Which keywords were used? Author keywords? Also, the explanation of the map is poor. I suggest adding a table with a list of the main keywords, their frequency and the cluster they belong to. In the main text, it is also  necessary to link the discussion of Figure 2  to the literature. It would be interesting to add the 'overlay map' built by using VOSViewer to identify the "historical" keywords and the keywords emerging in the recent years. This map will be discussed.”

Thank you very much for your recommendation based on your suggestion, we have introduced a table with frequencies and cluster membership, we have introduced the overlay analysis of the keywords and we have linked the discussion to the literature.

“Figure 1 (pag.12). Which research interests do the authors belonging to the four main clusters share?”

Thank you very much for your comment, based on your suggestion we have introduced the convergence points of the authors representing the main clusters identified.

5. Conclusions: The conclusions are just a summary of the findings. It is necessary to add some personal deductions and comments which also concern suggestions for the scientific community, policymakers and stakeholders. For example, which could be the future directions of the research? Which research lines should be strengthened?”

Thank you very much for your comment, based on your suggestion we have expanded the conclusions to include why it is important to review the literature on SDGs in organisations, as well as future lines of research on this topic. In this way, the revised version adds more value to the manuscript. Thank you very much for your feedback again as it has certainly helped us to improve the quality of the research.

Reviewer 3 Report

Dear Authors,

The manuscript is well written and could address all concerns regarding the research topic and problems. 

1. I strongly advise to reduce the conclusion volume and remove the repeated sentences.

2. unless your content's source is not you, mention the source. Otherwise it is not necessary to mention own source.

there are more comments on the attached pdf file.

Overall, manuscript is in good condition and I would suggest a minor revision.

Thank you.

Author Response

“ 1. I strongly advise to reduce the conclusion volume and remove the repeated sentences.”

Thank you very much for your recommendation, based on your recommendation we have cut some parts of the conclusions. However, we have had to introduce several aspects suggested by other reviewers. Also, repetitive and monotonous sentences have been removed. Thank you very much for your comments as they have certainly helped us to improve the clarity and quality of the study.

“2. unless your content's source is not you, mention the source. Otherwise it is not necessary to mention own source.”

Thank you very much for your advice, based on your recommendation we have removed the sources of the tables and figures entered in the results (since in all of them we are the sources ourselves).

there are more comments on the attached pdf file.

Thank you very much for your comments, we have tried to improve all the aspects shown in the attached pdf.

“Overall, manuscript is in good condition and I would suggest a minor revision. Thank you.”

Thank you very much for your efforts in reviewing the research.